# Underwater Sea Cucumber Identification Based on Improved YOLOv5

**Xianyi Zhai** **, Honglei Wei \*, Yuyang He, Yetong Shang and Chenghao Liu**

School of Mechanical Engineering and Automation, Dalian Polytechnic University, Dalian 116034, China
\* Correspondence: weihl2005@163.com

**Abstract:** In order to develop an underwater sea cucumber collecting robot, it is necessary to use the machine vision method to realize sea cucumber recognition and location. An identification and location method of underwater sea cucumber based on improved You Only Look Once version 5 (YOLOv5) is proposed. Due to the low contrast between sea cucumbers and the underwater environment, the Multi-Scale Retinex with Color Restoration (MSRCR) algorithm was introduced to process the images to enhance the contrast. In order to improve the recognition precision and efficiency, the Convolutional Block Attention Module (CBAM) is added. In order to make small target recognition more precise, the Detect layer was added to the Head network of YOLOv5s. The improved YOLOv5s model and YOLOv5s, YOLOv4, and Faster-RCNN identified the same image set; the experimental results show improved YOLOv5 recognition precision level and confidence level, especially for small target recognition, which is excellent and better than other models. Compared to the other three models, the improved YOLOv5s has higher precision and detection time. Compared with the YOLOv5s, the precision and recall rate of the improved YOLOv5s model are improved by 9% and 11.5%, respectively.

**Keywords:** YOLOv5; sea cucumber identification; object detection; deep learning; computer vision; models

## 1. Introduction

With the improvement of living standards, people's demand for sea cucumbers and other sea treasures is increasing. Sea cucumbers have been considered health foods with extremely high nutritional value since ancient times and are widely used in the food and pharmaceutical industries. Sea cucumbers are mainly raised in captivity; bottom sowing culture is the main culture method, that is, sea cucumbers seedlings are sown on the seabed and harvested after they are grown. Due to the special growth environment of sea cucumbers, mainly relying on artificial diving for fishing, the harvesting process is dangerous, and the return is low, so aquaculture enterprises need automated equipment to replace manual operations [1]. The key point in the development of underwater automation equipment is the need to identify and fish underwater targets. The color of sea cucumbers and the color of the environment are closer, and the degree of differentiation is lower. In order to be able to capture sea cucumbers more precisely in the underwater environment, the research on the precise and rapid identification of sea cucumbers is of great value to the development of sea cucumber harvesting robots and the realization of automated underwater fishing.

In the study of sea cucumber identification, many scholars have conducted research and exploration. Li Juan et al. conducted research based on computer vision technology [2], conducted edge detection on underwater sea cucumber, extracted the centroid of sea cucumber thorns, and identified underwater sea cucumber by oval fit, with a precision of 93.33%. With the rapid development of related disciplines such as machine learning, it is also gradually being used in the field of sea cucumber identification. Qiang Wei et al. proposed an underwater object detection algorithm based on improved Single Shot

MultiBox Detector (SSD) [3], and the detection precision of this method reached 95.06%. Lin Ning et al. proposed underwater target recognition detection based on Cascade Region with CNN features (RCNN) [4], which effectively improved the recognition and detection effect of small targets by adjusting hyperparameters, with an average precision of 89.1%. Ma Kongwei proposed the Convolutional Neural Network (CNN)-based sea cucumber identification technology and its application in underwater robots [5], and combined the lightweight model MobileNet with the SSD model; the average precision reached 90.06%, the data storage method was optimized, and the model size was reduced by 75%. In the study of the underwater object detection algorithm based on Faster-RCNN proposed by Wang Lu et al., the deeper neural network ResNet-101 was replaced with the original VGG-16 network [6], and the average test precision of the dataset reached 63.03%.

The above method uses the morphological characteristics, appearance characteristics, and characteristics of the sea cucumber that are different from the environmental color or combines multiple features to identify the sea cucumber. However, the vast majority of algorithms are aimed at the laboratory environment; because of the influence of aquatic plants, other marine animals and human garbage, there are too many uncertainties in the actual underwater environment in which the sea cucumber is located. Sea cucumbers are gregarious, and once there is overlap and occlusion, the recognition precision will be greatly reduced.

In 2015, Joseph Redmon proposed the One-stage algorithm YOLOv1 [7]. RCNN as a classic object detection method, has an essential difference from YOLO in that the detection process of RCNN is divided into two steps: classification problem and regression problem. YOLO only operates on one step regression problem, so the YOLO algorithm processes images faster; YOLO uses full-graph information for training and prediction to reduce the error rate of prediction, and since the RCNN algorithm cannot take advantage of the entire image, plaques in the background may be mistakenly identified as targets during detection. Xu Jianhua et al. proposed an underwater target recognition and tracking method based on YOLOv3 algorithm [7], which realized the identification, positioning, and tracking of underwater targets, with an average precision of 75.1% and a detection speed of 15 frames per second. Zhang Conghui proposed a sea cucumber target recognition algorithm based on tiny-YOLOv3 network [8], which modified and optimized the original YOLO; the detection precision reached more than 90%, and it could still reach 47 frames per second on the low-performance computing platform. Zhu Shiwei et al. proposed underwater object detection based on class-weighted YOLO network [9], with an average precision of 78.3% and a processing speed of 12 frames per second, which greatly improved the multi-target recognition effect. Therefore, the YOLO algorithm takes into account the characteristics of high precision and small memory occupation.

Aiming at the problems of low degree of differentiation of sea participation environment in natural environment and imprecise identification of small targets, an underwater sea cucumber recognition algorithm based on improved YOLOv5 is proposed.

(1) Improve the structure of the YOLOv5s model, increase the number of upsampled in the Neck network, and increase the Detect layer in the Head network, so that small sea cucumber targets can be detected;

(2) Add the CBAM module, which can save parameters and computing power, and ensure that it can be integrated into the existing network architecture as a plug-and-play module;

(3) The image was pre-processed by MSRCR algorithm, which enhanced the differentiation of the sea participation environment and provided help for precise and rapid identification of sea cucumbers in the natural environment.

(4) After modifying the YOLOv5s model, ablation study was conducted, and the feasibility of improvement was proved. Compared with YOLOv4 and Faster-RCNN, the experimental results showed that the improved YOLOv5s had a higher precision and recall rate.

In Section 2, the images are processed using MSRCR and the dataset is augmented with OpenCV. In Section 3, an introduction to the YOLOv5s network structure is introduced, the YOLOv5s network structure is improved, the CBAM and the Detect layer of the Head network are added, and the Ablation Study is conducted. in Section 4, the dataset using the improved YOLOv5s model is trained and the training results are analyzed. In Section 5, YOLOv5s, YOLOv4, and Faster-RCNN are compared using an improved YOLOv5s model and the experimental results are analyzed. Finally, conclusions and outlooks are provided in Section 6.

## 2. Materials and Methods

The underwater sea cucumber identification model training and detection process is shown in Figure 1. First, the images are processed by MSRCR algorithm, then the image is labeled by the labeling tool, the TXT format label is exported after the labeling is completed, and the image data of the label is amplified at the same time as the corresponding TXT file. Finally, YOLOv5s is used to train the images and obtain the model and test model performance.

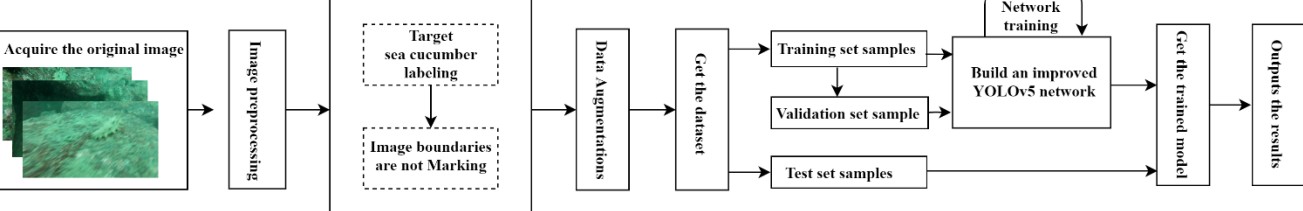

**Figure 1.** Sea cucumber identification flowchart.

### 2.1. Experimental Data Acquisition

Sea cucumber images were recorded by divers from the sea cucumber farming waters of Zhangzi Island. Recording time is 19 May 2021 9:00–11:00 a.m. Collected under natural light conditions during the day, when collecting, divers use a high-definition digital camera to record a distance of 30 to 70 cm from the sea cucumber and take a screenshot of the original image from the video; the resolution of the picture is $640 \times 360$, a total of 892 images were obtained, and some of the sea cucumber images are shown in Figure 2.

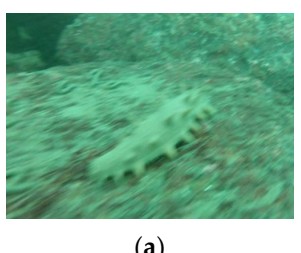 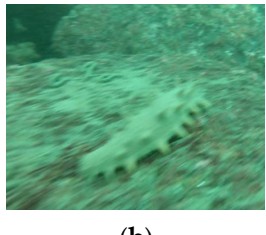 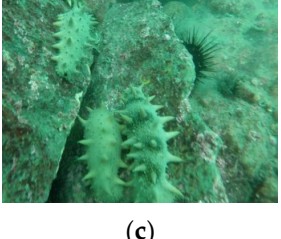 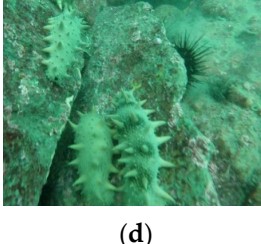

(**a**)                  (**b**)                  (**c**)                  (**d**)

**Figure 2.** (**a**–**d**) Partial sea cucumber images.

### 2.2. Image Preprocessing

Since the images collected underwater are too low in brightness, distortion, and difficult to distinguish between sea participation backgrounds, which makes it difficult to train the model, MSRCR is introduced to preprocess the images to enhance the sensitivity of sea participation backgrounds, as shown in Formula (1) [10,11].

$$R_{MSRCR_i}(x,y) = C_i(x,y)R_{MSR_i}(x,y) \tag{1}$$

Thereinto, $R_{MSRCR_i}(x,y)$ represents the image recovered by the $i$ th channel after using the algorithm; $C(x,y)$ is the color recovery factor, which is used to adjust the color distortion caused by local contrast enhancement, calculated as shown in Equation (2), where $\alpha$ is the regulator factor, $\beta$ is the gain constant, and $I(x,y)$ represents the original image; $R_{MSR_i}(x,y)$

shows the recovery image of the *i* th channel after using multiscale filtering, calculated as shown in Equation (3), thereinto $G_n(x, y)$ is a single-scale Gaussian filter, $\lambda$ represents the weight, *m* is the number of scales, and the value taken in this article is 3 [12–14].

$$C_i(x, y) = \beta \left\{ \log[\alpha \cdot I_i(x, y)] - \log\left[\sum_{i=1}^{N} I_i(x, y)\right] \right\} \tag{2}$$

$$R_{MSR_i}(x, y) = \sum_{n=1}^{m} \lambda_n \{\log I(x, y) - \log[G_n(x, y) \cdot I(x, y)]\} \tag{3}$$

As shown in Figure 3, the contrast ratio before and after the MSRCR processing has been significantly enhanced.

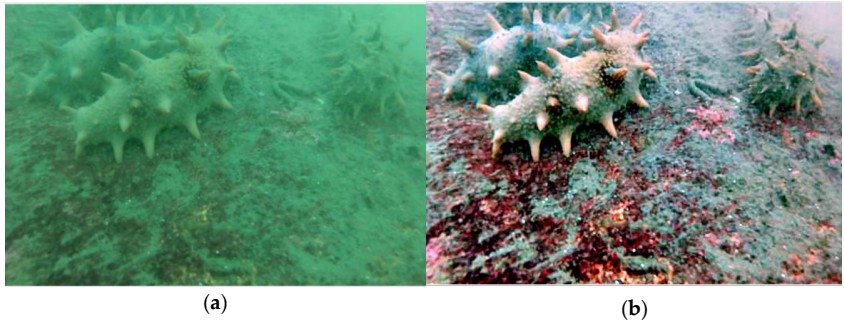

(a)          (b)

**Figure 3.** Image (**a**) is the original image. Image (**b**) is processed by MSRCR algorithm.

### 2.3. Data Augmentation

In the process of deep learning training, the number of training pictures has a certain impact on the model. Due to the limited number of sea cucumber images collected, data augmentation is required prior to training [15]. In the actual use of sea cucumber robot harvesting process, due to light, distance, angle, and water flow caused by jitter, there are often angle, position, size, brightness changes, and blurry images. In order to provide more comprehensive training images, using the data augmentation module included with YOLOv5 and combined with OpenCV under the Pytorch framework, the collected sea cucumber images are mirrored; the positive and negative angles are rotated by 45°, translated, randomly scaled, and randomly cropped; brightness changes and other operations are expanded; and a total of 5017 images to be trained are obtained. Some of the images are shown in Figure 4.

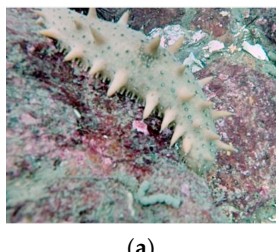 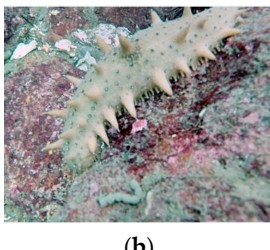 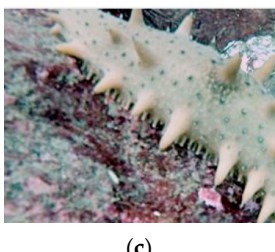 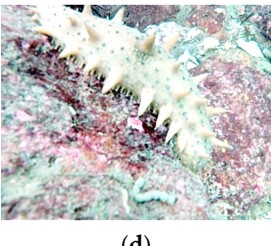

(a)        (b)        (c)        (d)

**Figure 4.** Image (**a**) is the original image, image (**b**) is after image flipping, image (**c**) is after randomly enlarged, and image (**d**) is after brightness change processing.

### 2.4. Dataset Preparation

The training images are divided into 6 groups of image libraries, using the online labeling website Make Sense, using rectangular boxes for sea cucumbers with complete morphology; sea cucumbers with occlusion or overlap, labeling the parts displayed in the image; and not labeling sea cucumbers that are located in the image border and display area less than 15%. After the tag was complete, the Visual Object Classes (VOC) type dataset was exported. Two sets of datasets were established, namely single image library

and mixed image library, and 80% of the images were selected as the training set, 10% as the validation set, and 10% as the test set.

## 3. Sea Cucumber Identification Network

In the latest version of the official code of YOLOv5, a total of five different depths and widths of YOLOv5n, YOLOv5s, YOLOv5m, YOLOv5l, and YOLOv5x are provided [15]. In this study, due to the small amount of sea parameters recognized and the high-speed requirements, the relatively small model YOLOv5s was selected as the training model.

### 3.1. YOLOv5 Network Model

The YOLOv5 network model is shown in Figure 5 and includes components such as Backbone, Neck, and Head. In the Backbone network of YOLOv5, the CSPDarknet53 structure is used as the basic network, and the main role of the Backbone network is to extract the basic characteristics of the input information [15,16]. The Neck network is located between the Head network and the Backbone network, and its main function is to further enhance the diversity of features to be extracted and the robustness of the network. The Head network function is the result of output detecting the target. The YOLOv5s model had three levels of direct, corresponding to three sets of initialized Anchor values. The Head network receives three feature maps of different scales, and then performs grid prediction on these three feature maps and uses convolution operation to obtain feature output [17]; and the detection was carried out by mainly identifying the three dimensions of large target, medium target, and small target [18].

Compared with the previous generation YOLOv4, the First Step was to add the Focus structure to prevent information loss when downsampling. The Focus module is a slice of the image before it enters the backbone. The specific operation is to get a value for every other pixel in a picture, similar to proximity downsampling, so that four pictures are obtained, and the four pictures are complementary, but no information is lost. In this way, the horizontal and vertical information is concentrated in the channel space, and the input channel is expanded by 4 times, that is, the stitched picture becomes 12 channels compared to the original RGB three-channel mode [19,20]. Finally, the resulting new image is convoluted to obtain a double downsampled feature map without information loss.

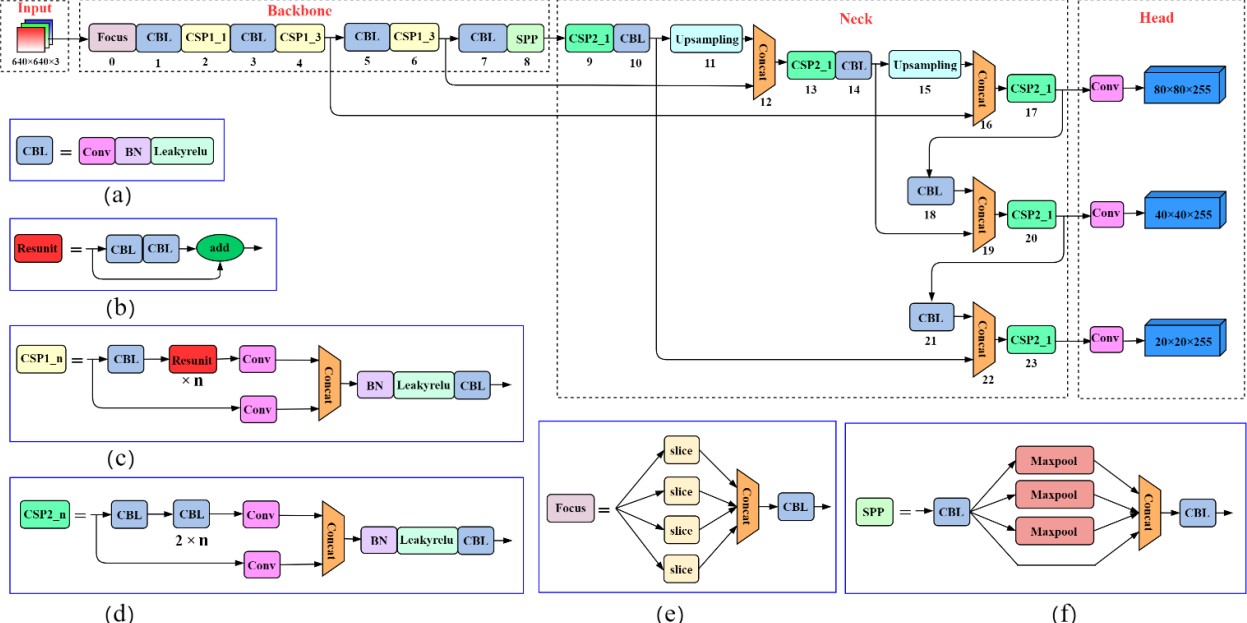

**Figure 5.** YOLOv5 network model: Image (**a**) is the CBL module, the CBL module consists of a Conv+BN+Leakyrelu activation function, Conv is convolution; image (**b**) Resunit borrowing from

the residual structure of the Resnet network, the network can be built deeper; image (**c**) CSP1_n, borrowing from the CSPNet network structure, which is composed of CBL module, Resunint module, Conv and Concate, and image (**d**) CSP2_n, borrowing from the CSPNet network structure, which consists of Conv and *n* Resunint modules [21], Concat; image (**e**) is the Focus structure, which first Concats multiple slice results, and then feeds them into CBL module; image (**f**) uses the maximum pooling method of $1 \times 1, 5 \times 5, 9 \times 9, 13 \times 13$ to perform multi-scale fusion [22].

Taking YOLOv5s as an example, the original $640 \times 640 \times 3$ image input Focus structure, using a slice operation, first becomes a feature map of $320 \times 320 \times 12$, and then after a convolution operation, it finally becomes a feature map of $320 \times 320 \times 32$ [23]. The slicing operation is shown in Figure 6.

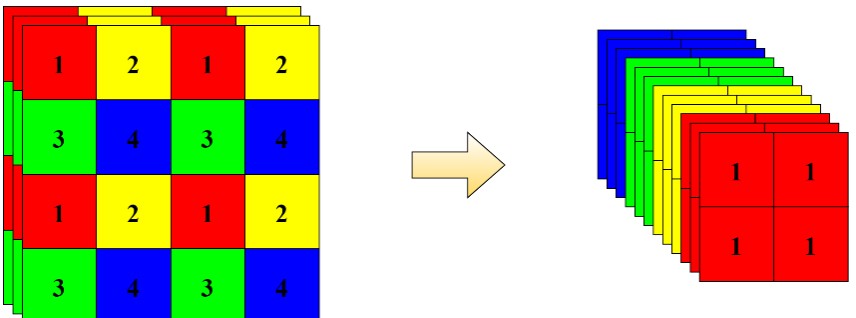

**Figure 6.** Focus structure slice operation.

### 3.2. Loss Function

The most commonly used calculation metric for bounding box regression loss is Intersection over Union (IoU) [17,24], as shown in Equation (4).

$$\text{IoU} = \frac{|A \cap B|}{|A \cup B|} \tag{4}$$

IoU algorithms are widely used because of scale invariance and non-negativity. However, since the IoU does not consider the distance between the real box and the prediction box, the direct use of the IoU as a Loss function will expose various problems: if there is no overlapping part between the prediction box and the real box, according to the definition, the IoU is 0, and the distance between the two boxes cannot be calculated. Due to the different alignment of different angles in different directions, the IoU will be exactly the same, and it is impossible to judge the coincidence between the prediction box and the real box. The YOLOv5 algorithm uses Generalized Intersection over Union_Loss (GIoU_Loss) as a loss function for bounding box regression [25,26]. GIoU pays attention to the overlapping parts as well as other non-overlapping parts, which can more precisely describe the various overlaps between the prediction box and the real box. Suppose *A* is the prediction box and *B* is the real box, so that *C* represents the smallest closed box containing *A* and *B*. The calculation formula (5) of the GIoU_Loss is as follows.

$$\text{GIoU\_Loss} = 1 - \text{GIoU} = \text{IoU} - \frac{|C/(A \cup B)|}{|C|} \tag{5}$$

### 3.3. YOLOv5 Network Improvements

Underwater sea cucumbers vary greatly in posture and size; the YOLOv5 is not ideal for the detection of small targets, and some sea cucumbers cannot be recognized and imprecisely identified in actual tests. In order to improve the detection ability of small objects, the YOLO structure was improved.

### 3.3.1. Add Convolutional Block Attention Module

In order to improve the efficiency of YOLO detection, the CBAM module was added. CBAM is based on feed-forward convolutional neural networks, through a given intermediate feature layer, first through a channel attention module, after obtaining the weighted result, then through a spatial attention module, and finally weighted to obtain the attention map. Multiply the attention graph by the input feature map for adaptive feature optimization; the structure shown in Figure 7. In this experiment, CBAM was added to the CBL structure, specifically behind The Rakyrelu, as shown in Figure 8a.

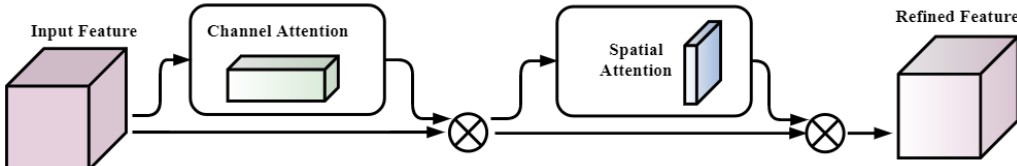

**Figure 7.** Convolutional Block Attention Module structure.

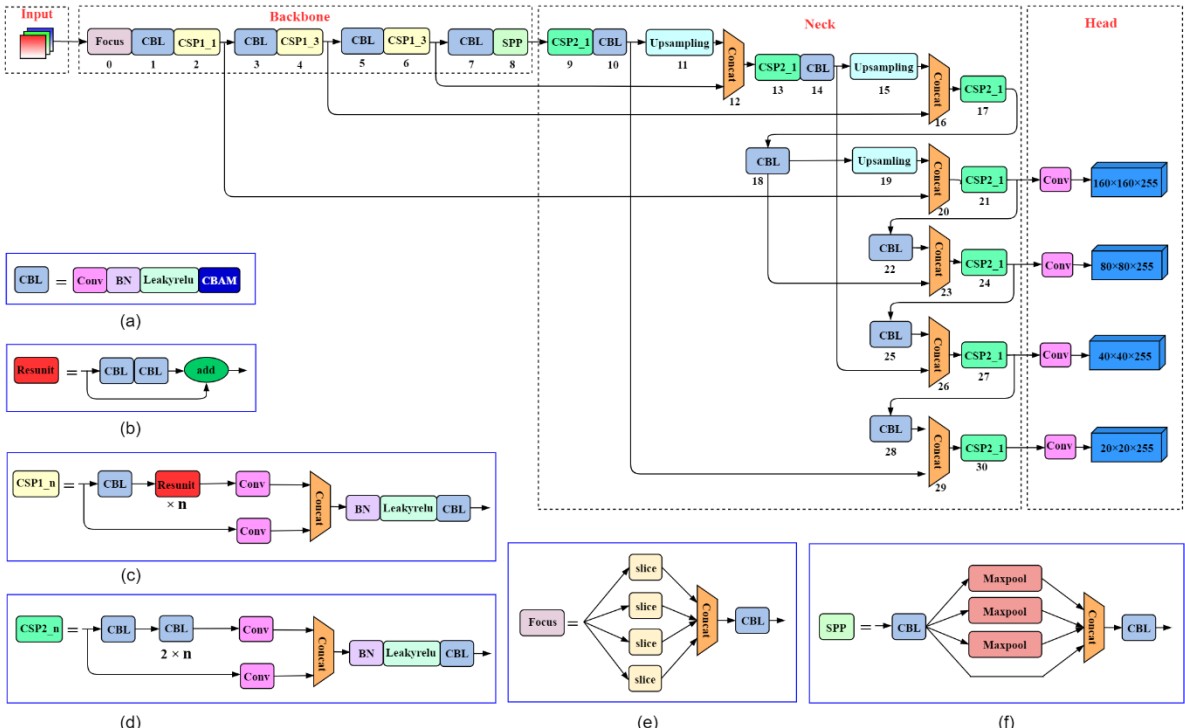

**Figure 8.** Improved YOLOv5 model: Image (**a**) is the CBL module+CBAM; image (**b**) Resunit borrowing from the residual structure of the Resnet network, the network can be built deeper; image (**c**) CSP1_n, borrowing from the CSPNet network structure, which is composed of CBL module, Resunint module, Conv and Concate, and image (**d**) CSP2_n, borrowing from the CSPNet network structure, which consists of Conv and *n* Resunint modules [21], Concat; image (**e**) is the Focus structure, which first Concats multiple slice results, and then feeds them into CBL module; image (**f**) uses the maximum pooling method of $1 \times 1, 5 \times 5, 9 \times 9, 13 \times 13$ to perform multi-scale fusion [22].

### 3.3.2. Add Detect Layer

The improved model is shown in Figure 8. After layer 17, continue to upsample the feature map to ensure that the feature map continues to expand, and at the 23rd layer, the feature map obtained with a size of $160 \times 160$ is Concat fused with the feature map of the second layer in the Backbone network, so as to obtain a larger feature map for small target detection. In the detect layer after layer 30, add a small object detection layer [11,27]. Since the improved model adds a detect layer, the test speed response is slightly reduced, but the

detection effect for small targets is significantly improved, and the parameter performance pair is shown in Table 1.

**Table 1.** Improved YOLOv5s model compared to YOLOv5s, YOLOv5 + CBAM, and YOLOv5s + detect.

| Models | Precision | Training Duration (Hour) | Weight (MB) | Detection Speed (ms/pic) |
|---|---|---|---|---|
| YOLOv5s | 83.6% | 4.4 | 13.5 | 17 |
| YOLOv5s + CBAM | 89.1% | 4.9 | 14.4 | 21 |
| YOLOv5s + detect | 87.4% | 5.0 | 14.8 | 23 |
| YOLOv5s + CBAM + detect | 92.9% | 5.5 | 15.6 | 25 |

### 3.3.3. Ablation Study

At the same time, an ablation study was conducted [28], using the improved YOLOv5s compared with YOLOv5s, YOLOv5s + CBAM, and YOLOv5s + detect. Select 1035 sea cucumber images as the dataset, the training set, test set and validation set ratio is 8:1:1, the same platform is used for training, the training epoch is set to 200, and the final result is shown in Table 1. The experimental results show that only the CBAM module is added to the YOLOv5s, only the detect layer is added, and the CBAM module and the detect are added at the same time; all three cases can improve the precision, but because the complexity of the YOLOv5s model increases, the training time will be longer, the weight generated by the training will increase, and the time to recognize the picture will increase slightly.

## 4. Model Training and Testing

### 4.1. Experimental Platform

The experimental environment of this study is shown in Table 2.

**Table 2.** Experimental environment.

| Configuration | Parameter |
|---|---|
| CPU | Intel Core i5-7300HQ |
| GPU | NVIDIA GeForce GTX1050Ti 4G |
| Operating system | Windows10 |
| Environment | Cuda11.2 |
| Development platform | PyCharm2021.3 |
| Others | OpenCV4.5.5, Numpy1.20.3 |

### 4.2. Model Testing

In order to avoid over-fitting, 10-Folder cross-validation method is adopted in the experiment, and it is shown in Figure 9.

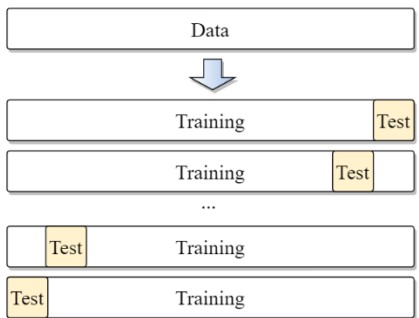

**Figure 9.** 10-Folder cross-validation.

The curve of the parameters with the number of epochs is shown in Figure 10. Figure 10a is the precision plot, that is, the correct positive class/all the positive classes found [29]; the precision calculation formula is shown in Equation (6), True Positives (TP) means the number of images that are predicted correctly and the predicted result is positive

(indicating that a positive sample is used), and False Positives (FP) means the number of pictures that are predicted incorrectly and the predicted result is positive (indicating that a negative sample is used) [30,31]. When the IoU is set to 0.5, the improved model precision shows a stable upward trend, and after 100 epochs, the precision of the training result reaches 99.6%.

$$precision = \frac{TP}{TP + FP} \tag{6}$$

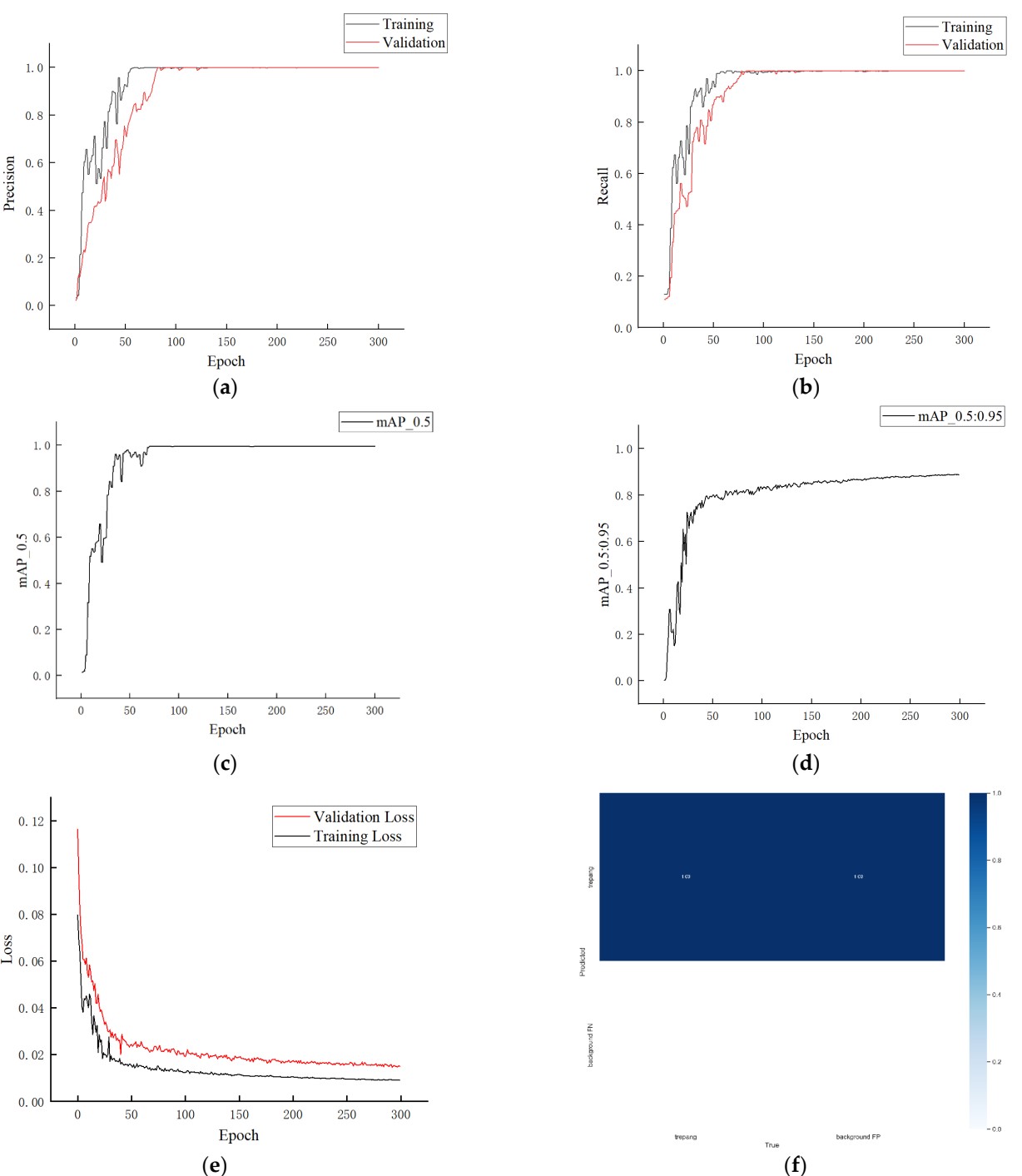

**Figure 10.** Curve of parameters with number of epochs: image (**a**) is the precision change curve; image (**b**) is the recall change curve; image (**c**) and image (**d**) are mAP curves; image (**e**) is the loss value change curve; and image (**f**) is the Confusion matrix.

Figure 10b is the recall rate diagram, that is, the positive class that should be found correctly/all the positive classes that should have been found; the recall calculation formula is shown in Equation (7) [29]. False Negatives (FN) means the number of pictures that are predicted incorrectly, and the predicted result is negative (indicating that a positive sample is being used) [30,31]. When the IoU is set to 0.5, the improved model shows a stable upward trend, and after 100 epochs, the precision of the training result reaches 99.5%.

$$recall = \frac{TP}{TP + FN} \tag{7}$$

The metric for measuring recognition precision in object detection is the minute average precision (mAP). In multiple categories of object detection, each category can draw a curve based on recall and precision [31], AP is the area under the curve, and mAP is the average of multiple category APs. The higher the value of the AP, the higher the average precision of the model; mAP_0.5 and mAP_0.5:0.95 are the two most commonly used metrics. Moreover, mAP_0.5 indicates when the IoU is set to 0.5, the APs of all images of each class are calculated [27], and then all the categories are averaged; mAP_0.5:0.95 indicates the average mAP on different IoU thresholds (from 0.5 to 0.95, step size 0.05, i.e., 0.5, 0.55, 0.6, 0.65, 0.7, 0.8, 0.85, 0.9, 0.95). Figure 10c represents mAP_0.5 curve, and Figure 10d represents mAP_0.5:0.95 curve.

In Figure 10e, the loss curves for the training and validation sets are shown; the smaller the value indicates that the more precise the box [28]. When the IoU is set to 0.5, with the increase of the training period, the loss curve of the training set and the verification set shows a downward trend and eventually tends to be stable; after 100 epochs, the loss of the training set reaches the lowest value of about 0.02, and the loss of the verification set is less than 0.02.

Figure 10f is the confusion matrix. Confusion matrix is a case analysis table summarizing the prediction results of classification model.

## 5. Results and Analysis

In order to verify the performance of the target recognition algorithm of underwater sea cucumber, the single image library and the mixed image library were trained and tested using the improved YOLOv5s, YOLOv5s, YOLOv4, and Faster-RCNN models.

### 5.1. Sea Cucumber Identification for the Single Gallery

Select a single image library taken in the same video and test it using the YOLO model and the improved model, respectively. There are 845 images in total, with training sets, validation sets, and test sets distributed according to 8:1:1. Figure 10 shows the comparison of the improved YOLOv5s, YOLOv5s, YOLOv4, and Faster-RCNN test results. Figure 11a,c are the results of YOLOv5s,;Figure 11b,d are the results of improved YOLOv5s; Figure 11e is the YOLOv4 result; and Figure 11f is the Faster-RCNN result. Figure 11a,b comparisons show that improved YOLO can detect when YOLOv5s does not detect small targets at the edge. Figure 11c,d comparisons show that when the shooting angle is changed, YOLOv5 can detect small sea cucumbers at the edge, and the improved YOLO detects a higher level of confidence in the target. YOLOv4 result shows that the target at the edge is not detected, and the prediction box deviation is large. Faster-RCNN result shows that the target at the edge is not detected, and the prediction box deviation is still there and needs to be optimized.

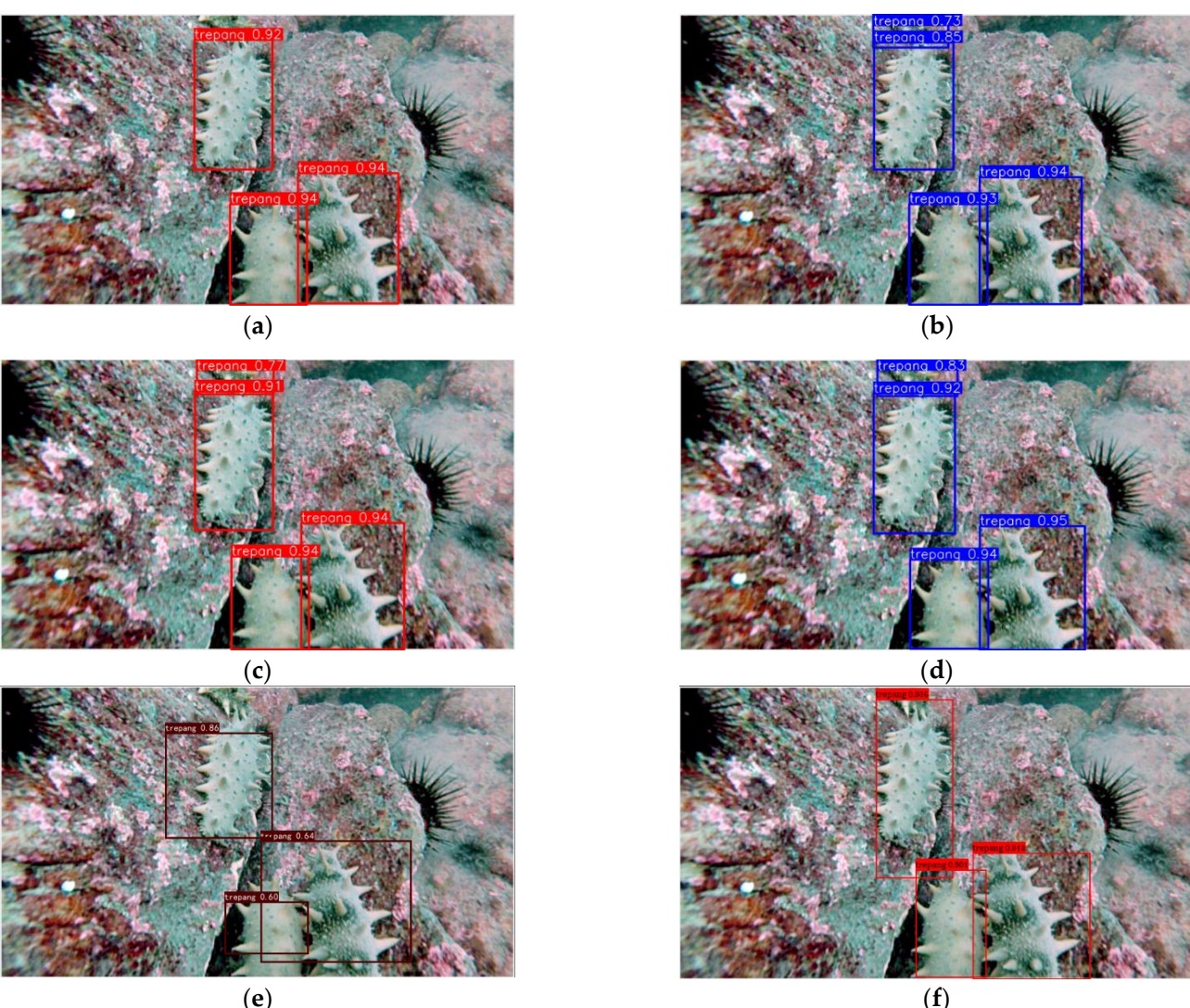

**Figure 11.** Test the Image After Improvements: image (**a**–**c**) is the results of the YOLOv5s; image (**b**–**d**) is the results of the improved YOLOv5s; image (**e**) is the result of YOLOv4; and image (**f**) is the result of Faster-RCNN.

### 5.2. Sea Cucumber Identification for Mixed Image Gallery

Take images of sea cucumbers from different angles and different forms from multiple videos and mix the images to obtain a mixed image library. There are 4172 images in total, with training sets, validation sets, and test sets distributed according to 8:1:1. Images of sea cucumbers were identified using YOLOv5s, improved YOLOv5s, YOLOv4, and Faster-RCNN, and the experimental results are shown in Figure 12. A1, B1, C1, D1, and E1 of Figure 12 are the YOLOv5s identification results; A2, B2, C2, D2, and E2 of Figure 12 are the improved YOLOv5s identification results; A3, B3, C3, D3, and E3 of Figure 12 are the YOLOv4 recognition results, and A4, B4, C4, D4, and E4 of Figure 12 are the Faster-RCNN identification results.

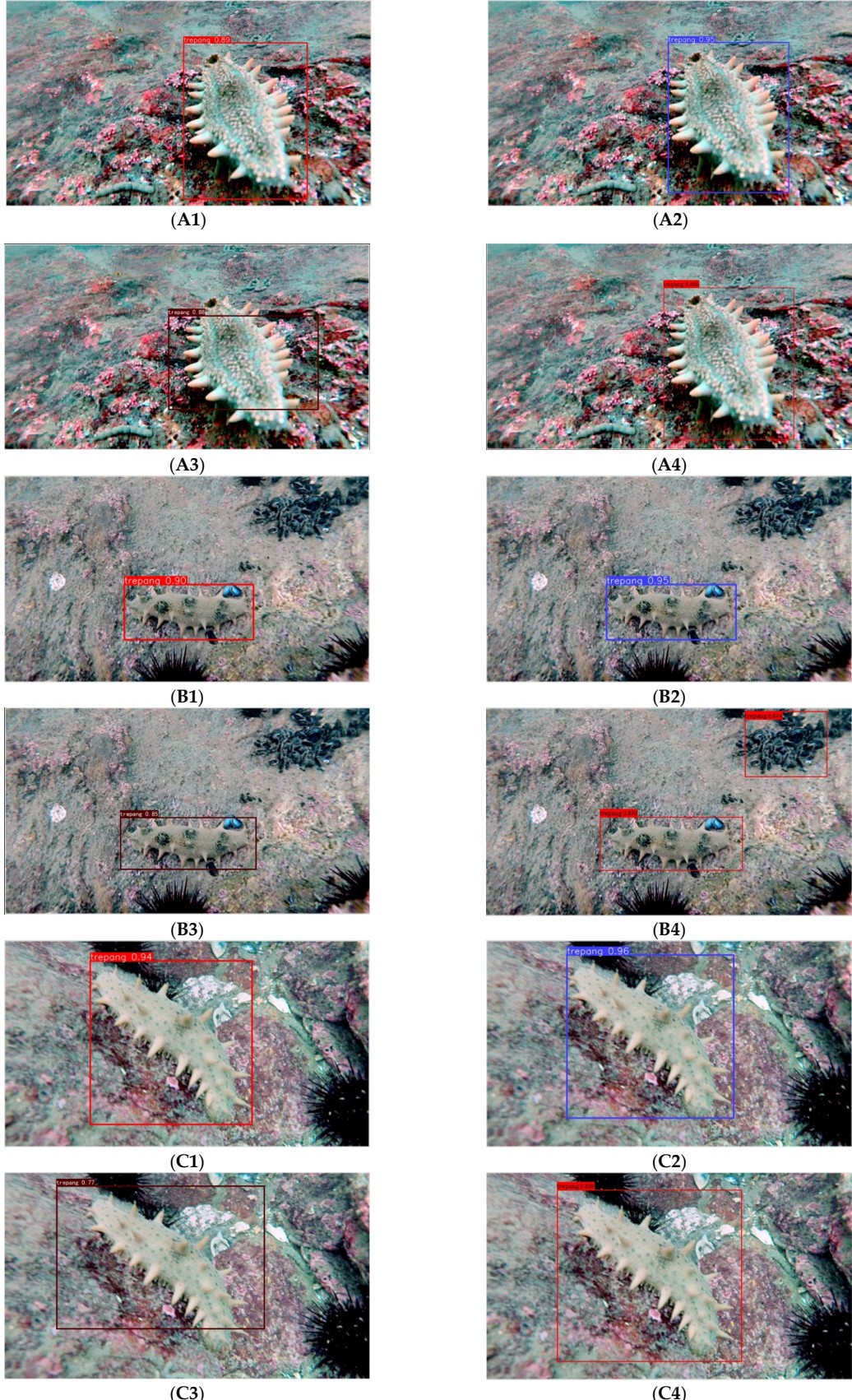

**Figure 12.** *Cont.*

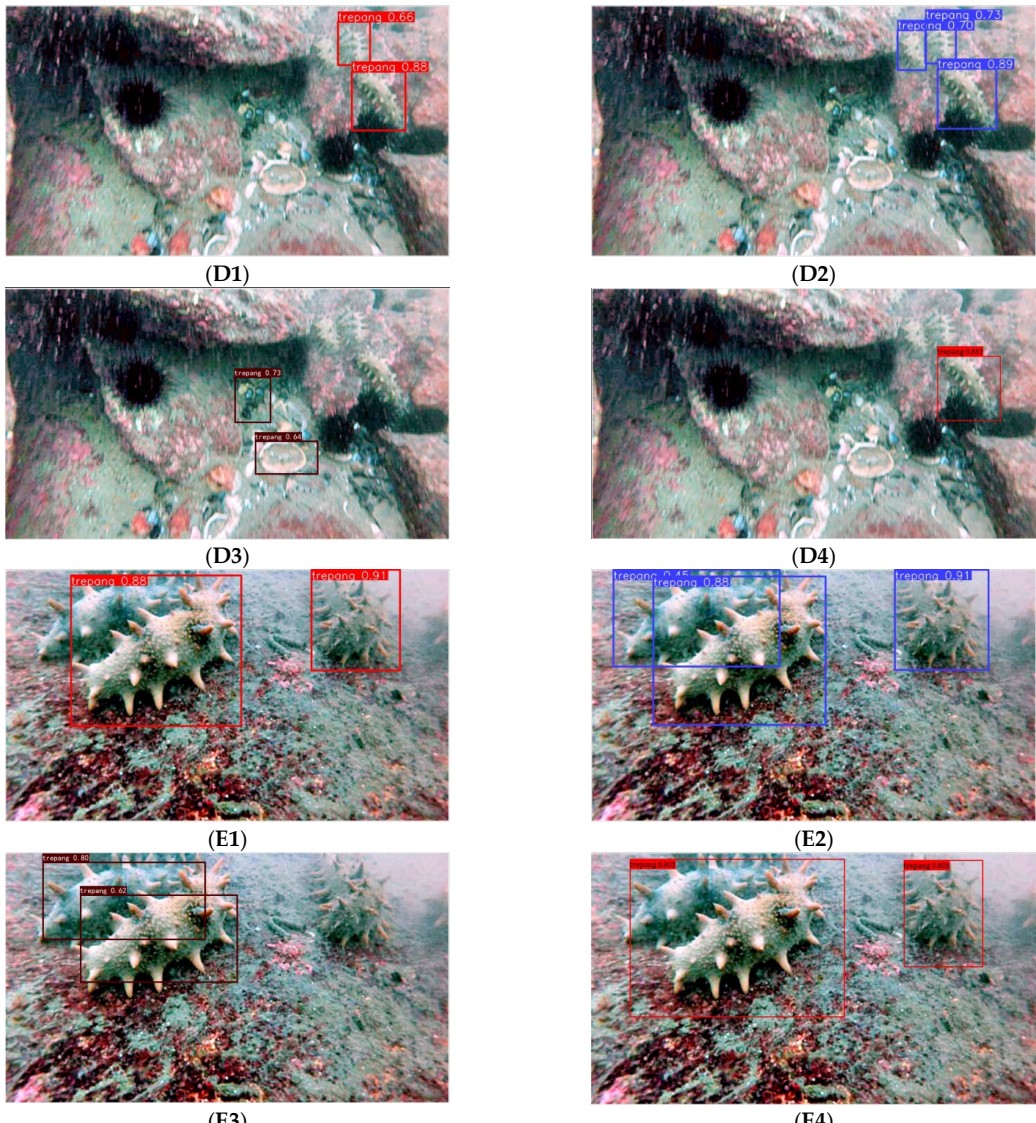

**Figure 12.** Images detection using improved YOLOv5s, YOLOv5s, YOLOv4, and Faster-RCNN. Image (**A1,B1,C1,D1,E1**) are the results of YOLOv5s; Image (**A2,B2,C2,D2,E2**) are the results of improved YOLOv5s; Image (**A3,B3,C3,D3,E3**) are the results of YOLOv4; (**A4,B4,C4,D4,E4**) are the results of Faster-RCNN.

It can be seen from the experimental results that the improved YOLOv5 model has a higher recognition effect on small targets in the image than the other three models, a more precise frame selection range for the prediction box, and a higher confidence in the same target. For example, in the YOLOv5s results, a missed detection is seen in Figure 12(D1). In the YOLOv4 results, there was an imprecise prediction box problem in Figure 12(A3), an error detection and missed detection in D3, and a missed test in E3. In the Faster-RCNN results, Figure 12(B4) has error detection and D4 and E4 have missed detection.

## 5.3. Comparison of Test Results

Table 3 compares the test results of the YOLOv5s and other three models when the IoU is set to 0.5. The improved YOLOv5s model has a precision increase of about 9%; a recall rate of about 11.5%; and a slight reduction in detection speed due to an increase in computational volume due to an improved method.

**Table 3.** Improved YOLOv5s model compared to YOLOv5s, YOLOv4, and faster-RCNN.

| Models | Precision | Recall | Detection Speed (ms/pic) |
|---|---|---|---|
| Improved YOLOv5s | 97.1% | 96.0% | 22 |
| YOLOv5s | 88.1% | 84.5% | 19 |
| YOLOv4 | 78.3% | 77.4% | 31 |
| Faster-RCNN | 82.4% | 80.9% | 25 |

## 6. Conclusions

A method based on improving YOLOv5 is proposed, with the aim of solving the problem of small target recognition, improving the recognition precision, and achieving precise identification of underwater sea cucumbers under natural conditions. The results of this paper are summarized as: (1) Using the MSRCR module, the image is processed to reduce distortion and enhance contrast, improve the sensitivity of the sea participation environment, reduce the difficulty of model training, and the effect of MSRCR algorithm on image optimization is more significant. (2) According to the actual situation of underwater sea cucumber, the YOLOv5 algorithm is selected as the basis for method implementation. In order to improve the recognition effect of small targets, the YOLOv5s model has been improved, the CBAM attention mechanism module has been added, and the addition of the CBAM module greatly improves the efficiency and accuracy of training at the expense of a small amount of detection time. The Head network has been improved, a layer of Detect layer has been added, Ablation Study was conducted, and the precision was improved, demonstrating the feasibility of the operation. The increased Detect layer carries out more upsampling on the input image and adds a detection layer for small targets, which makes the improved YOLOv5s model achieve further recognition precision for small targets. (3) Compared with YOLOv5s, YOLOv4, and Faster-RCNN, experimental results show that the detection effect of small targets has been significantly improved. Comparing with the YOLOv5s, when IoU = 0.5, the precision rate increased by 9%, the recall rate increased by 11.5%, for small targets that YOLOv5s cannot identify, and the improved model can easily identify them with a high confidence. Confidence in other targets also improved, so the small target recognition effect was significantly improved, and the improved method had significant advantages.

**Author Contributions:** H.W. and X.Z. proposed the idea of the paper. X.Z. and Y.H. helped manage the annotation group and helped clean the raw annotation. X.Z. conducted all experiments and wrote the manuscript. X.Z., C.L. and Y.S. revised and improved the text. H.W. and X.Z. are the people in charge of this project. All authors have read and agreed to the published version of the manuscript.

**Funding:** This research was funded by [Liaoning Provincial Department of Education 2021 annual Scientific research funding project] grant number [LJKZ0535, LJKZ0526] and [Comprehensive reform of undergraduate education teaching in 2021] grant number [JGLX2021020, JCLX2021008].

**Institutional Review Board Statement:** Not applicable.

**Informed Consent Statement:** Not applicable.

**Data Availability Statement:** Data that support the findings of this study are available from the corresponding author upon reasonable request.

**Conflicts of Interest:** The authors declare no conflict of interest.

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
