# Peer review of "Underwater Sea Cucumber Identification Based on Improved YOLOv5"

_applsci, doi:10.3390/app12189105_

Round 1
Reviewer 1 Report
Herewith attached all suggestions and comments to be implemented in the manuscript as a pdf file.

Author Response
Dear Reviewers:
The authors are grateful to the reviewers for their time and effort on the manuscript. We have tried our best to improve and made some changes in the manuscript.
According to the suggestions of the reviewers, the author (1) changed the grammar errors and spelling errors, the format of the reference, the manuscript structure has been rearranged,(2) added the ablation study of the improved model, (3) increased the precision and recall curve of the validation set, added the mAP curves, (4) and added the comparison of the improved YOLOv5s with YOLOv4 and Faster-RCNN. Please see the attachment for the specific reply. However, there may still be some mistakes that the authors have not found, and I hope the reviewers will give me any comments.
Thanks again for the reviewer's efforts!
Kind regards,
Mr. Zhai

Reviewer 2 Report
The authors presented an article that claims to modify the YOLOv5s model thus enhancing its predictive ability of small-sized sea cucumbers.
1. First of all the English quality of the whole article needs improvement along with fixing typos (ex, "yolOv3" on line 66 and "Refineed Feature" in figure 6)
2. Line 58, where are the references to the YOLO publication and the information stated in your article?
3. Lines 77-84 need to be clearly written in bullets to better show the paper's contributions. Also, you need to introduce the upcoming sections
4. The order of the "materials and methods" section is very confusing, you shouldn't start by explaining the dataset collection. The section needs to start by explaining the whole proposed model ( which is shown in figure 8) and then explain each of the blocks in figure 8 in the following sections including the proposed YOLOv5s improvement. Kindly reorder this section.
5. Commenting on the dataset collection, further details of the cameras used to collect the dataset and specifically the resolution of the collected images are required. Also, the claim that the proposed model is for small image detection should be further illustrated by showing the size of the sea cucumber in the original image. Do the ones shown in Figure 1 show their actual size as collected by the camera?? If Yes, then they are not small!!
6. Line 136, there are 5 depths of YOLOv5, you missed YOLOv5n plus you need to reference the YOLOv5 page for the information you just mentioned "https://github.com/ultralytics/yolov5"
7. Line 144-145 What is meant by the benchmark network???
8. Line 147 refers to the head network, yet figures 4 and 7 name it the prediction. Please unify the terminology.
9. Figure 4 needs further explanation of all of its components.
10. Figures 4 and 7: You need to mention the specific size of the output of the layers.
11. Figures 4 and 7: You also need to show how the three outputs of the head are combined to give one decision
12. There is a small box in the lower middle of both figures 4 and 7 (starting with a focus layer) that I cannot relate to the remaining figure. Is this further explanation of layers 0 and 1??? If so, it needs to be better illustrated.
13. Line 189: where exactly is the CBAM added? you need to refer to figure 4 and mention after what layer exactly was it added
14. Line 199: You mentioned: "After Layer 17" but you didn't specify in which figure. If you are referring to figure 4 then it is very weird that you are continuing your explanation by referring to layer 30 which doesn't exist in figure 4. At the same time if you are referring to figure 7, then referring to layer 20 after it is confusing as it is a concat layer. So please properly explain the improvement and refer to the figures.
15. Line 221, You are not referring to the correct figure
16. Line 222, you refer to the accuracy plot, and I see a precision plot. Those are two completely different plots! PLease update.
17. Figure 9. You need to add the validation curves for both the precision and recall.
18. Line 242: Please unify your reference to the YOLOv5s algorithm throughout the article, don't name it " the before model" or "the original model"
19. In figure 10, it is not clear what is the difference between a and c if it is exactly the same image applied to the YOLOv5s. The same applies to b and c. Please add further explanation.
20. It is not clear why you need to have a single gallery (same video) test. Please explain what addition this test provides. If you want to have a test for a whole video following a specific sea cucumber then this should be done for all separate videos and averaged to avoid choosing a video that is too good or too bad. Again if so, specific details of the videos need to be provided as to how long the video is, the number of frames addressed.. etc.
21. You mentioned having two datasets extracted, therefore you need to mention the details (count of images, splits) of every single image and the mixed image libraries.
22. Given that the datasets are of very small sizes, you need to have cross-validation experiments to show that the model works properly on different portions of the datasets.
23. The results need further metrics. YOLO models are always assessed using the Mean Average Precision so you need to add it.
24. Also You need to add the confusion matrices of the results
25. You need to add the accuracy measure to your results as you referred to it multiple times yet it is not shown.
26. How does having multiple sea cucumbers in the same image affect the classification?
27. You need to show what exactly is meant by the positive class (line 222) and the negative class and how having multiple sea cucumbers affect that classification.
28. The conclusion needs to be written as paragraphs and updated to reflect the paper modifications.
Author Response

(The authors gave the same response as above.)

Reviewer 3 Report
Underwater target detection plays an important role in ocean exploration, to which the improvement of relevant technology is of much practical significance. For the better identification and efficient location of the underwater sea cucumber, the authors have proposed an improved method namely YOLOv5 .The results of the paper seem to be quite original, however, the same are lacking a proper strength. I suggest the authors to revise the manuscript by incorporating the following suggestions:
1. The abstract is too short to convey the purpose of the article eloquently.
2. There a several grammatical errors throughout the manuscript and due care should be taken while revising the manuscript.
3. The problem under investigation has been studied by other efficient and powerful methods including the deep leaning method, SSD algorithms, Principal component analysis and support vector machine methods. Surprisingly, a much recent article entitled “Underwater Target Detection Algorithm Based on Improved YOLOv5” J. Mar. Sci. Eng. 2022, 10, 310. https://doi.org/10.3390/jmse10030310 has not been cited in the manuscript, although most of the ideas have been borrowed from the paper.
4. Authors are suggested to explain in brief the major differences of their proposed method with the aforementioned method. In fact, authors are requested to provide comparisons of their results with the other existing methods like Principle Component and Deep learning methods
5. The reference list should be updated with few more recent works closely related to the undertaken problem. For instance,
a. Sea Cucumber Detection Algorithm Based on Deep Learning, Sensors 2022, 22, 5717. https://doi.org/10.3390/s22155717
b. Underwater sea cucumber identification based on Principal 1 Component Analysis and Support Vector Machine, https://www.sciencedirect.com/science/article/pii/S026322411830976X
c. “Underwater Target Detection Algorithm Based on Improved YOLOv5” J. Mar. Sci. Eng. 2022, 10, 310. https://doi.org/10.3390/jmse10030310

Author Response

(The authors gave the same response as above.)

Round 2
Reviewer 2 Report
The manuscript has vastly improved
1- The Bullets at the end of the introduction provide more of a summary but it should give a description of the unique contributions of the work
2- You still missed the = sign in part of figure 5e (focus)
3- Referring again to point 26 in the previous response. Can you further illustrate if, for example, one image had 3 sea cucumbers of which one or two were correctly identified and the remaining were not, how will this affect the TP and FP? This is why I requested the confusion matrix as it will show if the total for the row is the number of images or the number of sea cucumbers. And again this needs further illustration.
4- I still believe you need to have a cross-validation experiment as even if the number of images is 5017, it is not large enough for a deep learning model.
5- I can still see lots of grammatical mistakes throughout the document
Author Response
Dear Reviewers:
The authors are grateful to the reviewers for their time and effort on the manuscript. According to the suggestions of the reviewers, we have tried our best to improve and made some changes in the manuscript again.Please see the attachment.
In addition, China's Teachers' Day and Mid-Autumn Festival are coming soon, and the Mid-Autumn Festival is a very important traditional festival in the eyes of Chinese, and the author wishes the reviewers family harmony! smooth work and good health!
Thanks again for the reviewer's efforts!
Kind regards,
Mr. Zhai

Reviewer 3 Report
The authors have incorporated all the suggested changes to the best of my satisfaction.
Author Response
Dear Reviewers:
The authors are grateful to the reviewers for their time and effort on the manuscript!
China's Teachers' Day and Mid-Autumn Festival are coming soon, and the Mid-Autumn Festival is a very important traditional festival in the eyes of Chinese, and the author wishes the reviewers family harmony! smooth work and good health!
Thanks again for the reviewer's efforts!
Kind regards,
Mr. Zhai